# Data-driven modeling of hydraulic head time series: results and lessons learned from the 2022 groundwater modeling challenge

Raoul A. Collenteur[1,*], Ezra Haaf[2], Mark Bakker[3], Tanja Liesch[4], Andreas Wunsch[5], Jenny Soonthornrangsan[27], Jeremy White[6], Nick Martin[7], Rui Hugman[6], Ed de Sousa[6], Didier Vanden Berghe[8], Xinyang Fan[9,10], Tim J. Peterson[11], Jānis Bikše[12], Antoine Di Ciacca[13], Xinyue Wang[14], Yang Zheng[14], Maximilian Nölscher[15], Julian Koch[16], Raphael Schneider[16], Nikolas Benavides Höglund[17], Sivarama Krishna Reddy Chidepudi[18,19], Abel Henriot[19], Nicolas Massei[18], Abderrahim Jardani[18], Max Gustav Rudolph[20], Amir Rouhani[21], J. Jaime Gómez-Hernández[22], Seifeddine Jomaa[21], Anna Pölz[23,24], Tim Franken[25], Morteza Behbooei[26], Jimmy Lin[26], and Rojin Meysami[26]

[1]Eawag, Department Water Resources and Drinking Water (W+T), Duebendorf, Switzerland
[2]Chalmers University of Technology, Department of Architecture and Civil Engineering, Gothenburg, Sweden
[3]Department of Water Management, Faculty of Civil Engineering and Geosciences, Delft University of Technology, The Netherlands
[4]Institute of Applied Geosciences, Division of Hydrogeology, Karlsruhe Institute of Technology, Germany
[5]Fraunhofer Institute of Optronics, System Technologies and Image Exploitation IOSB, Karlsruhe, Germany
[6]Intera, Fort Collins, Colorado, USA
[7]Southwest Research Institute (SWRI), San Antonio, Texas, USA
[8]Burgeap, Ginger Group, France
[9]Department of Geography and Geosciences, GeoZentrum Nordbayern, Friedrich-Alexander-University Erlangen-Nuremberg (FAU), Germany
[10]Institute of Geography & Oeschger Center for Climate Change Research, University of Bern, Switzerland
[11]Department of Civil Engineering, Monash University, Australia
[12]University of Latvia, Department of Geology, Latvia
[13]Environmental Research, Lincoln Agritech Ltd, Lincoln, New Zealand
[14]Data Science Institute (DSI), Brown University, Rhode Island, USA
[15]German Federal Institute for Geoscience and Resources (BGR)
[16]Geological Survey of Denmark and Greenland (GEUS), Department of Hydrology, Copenhagen, Denmark
[17]Department of Geology, Lund University, Sweden
[18]Univ Rouen Normandie, UNICAEN, CNRS, M2C UMR 6143, F-76000 Rouen, France
[19]BRGM, 3 av. C. Guillemin, 45060 Orleans Cedex 02, France
[20]Institute of Groundwater Management, Dresden University of Technology, Dresden, Germany
[21]Department of Aquatic Ecosystem Analysis and Management, Helmholtz Centre for Environmental Research - UFZ, Magdeburg, Germany
[22]Institute for Water and Environmental Engineering, Universitat Politècnica de València, Valencia, Spain
[23]Institute of Hydraulic Engineering and Water Resources Management, TU Wien, Vienna, Austria
[24]Interuniversity Cooperation Centre Water and Health, Austria
[25]Sumaqua, Louvain, Belgium
[26]University of Waterloo, ON, Canada
[27]Department of Geoscience & Engineering, Faculty of Civil Engineering and Geosciences, Delft University of Technology, The Netherlands

**Correspondence:** *Raoul Collenteur (raoul.collenteur@eawag.ch)

**Abstract.**

This paper presents the results of the *2022 groundwater time series modeling challenge*, where 15 teams from different institutes applied various data-driven models to simulate hydraulic head time series at four monitoring wells. Three of the wells were located in Europe and one in the USA, in different hydrogeological settings in temperate, continental, or subarctic climates. Participants were provided with approximately 15 years of measured heads at (almost) regular time intervals and daily measurements of weather data starting some 10 years prior to the first head measurements and extending around 5 years after the last head measurement. The participants were asked to simulate the measured heads (the calibration period), provide a prediction for around 5 years after the last measurement (the validation period for which weather data was provided but not head measurements), and to include an uncertainty estimate. Three different groups of models were identified among the submissions: lumped-parameter models (3 teams), machine learning models (4 teams), and deep learning models (8 teams). Lumped-parameter models apply relatively simple response functions with few parameters, while the artificial intelligence models used models of varying complexity, generally with more parameters and more input, including input engineered from the provided data (e.g., multi-day averages).

The models were evaluated on their performance to simulate the heads in the calibration period and to predict the heads in the validation period. Different metrics were used to assess performance, including metrics for average relative fit, average absolute fit, fit of extreme (high or low) heads, and the coverage of the uncertainty interval. For all wells, reasonable performance was obtained by at least one team from each of the three groups. However, the performance was not consistent across submissions within each group, which implies that application of each method to individual sites requires significant effort and experience. Especially estimates of the uncertainty interval varied widely between teams, although some teams submitted confidence intervals rather than prediction intervals. There was not one team, let alone one method, that performed best for all wells and all performance metrics. Four of the main takeaways from the model comparison are that: (1) Lumped-parameter models generally performed as well as artificial intelligence models, which means they capture the fundamental behavior of the system with only a few parameters. (2) Artificial intelligence models were able to simulate extremes beyond the observed conditions, which is contrary to some persistent belief about these methods. (3) No overfitting was observed in any of the models, also the models with many parameters, as performance in the validation period was generally only a bit lower than in the calibration period, which is evidence of appropriate application of the different models. (4) The presented simulations are the combined results of the applied method and the choices made by the modeler(s), which was especially visible in the performance range of the deep learning methods; underperformance does not necessarily reflect deficiencies of any of the models. In conclusion, the challenge was a successful initiative to compare different models and learn from each other. Future challenges are needed to investigate, e.g., the performance of models in more variable climatic settings, to simulate head series with significant gaps, or to estimate the effect of drought periods.

# 1 Introduction

Time series of hydraulic heads are one of the most important sources of information about groundwater systems. These time series contain information about the subsurface conditions and about the stresses causing the observed fluctuations. Modeling makes such information explicit and increases our understanding of groundwater systems (Shapiro and Day-Lewis, 2022). Modeling is essential to assess the impact of future land use and climatic scenarios on groundwater systems. Although the solution of groundwater-related problems often requires a spatial model, Bakker and Schaars (2019) argue that many problems can be solved by modeling the heads with a point-scale model (i.e., a time series model at a single monitoring well). Over the years, many types of models have been developed to simulate heads measured in a monitoring well. These models range from artificial intelligence to purely statistical models, and from simple analytic solutions to complex numerical (3D) models based on physical laws. The choice of a useful model can be challenging due to the wide range of available models for similar tasks. The choice of an appropriate model for a certain task is commonly based on the purpose of the model, data availability, and often on previous experience of the modeler (e.g., Addor and Melsen, 2019).

Studies that systematically compare different models can help in model selection. Comparing the performance of different models can help both practitioners and developers to improve existing models by learning from other modeling concepts (Kollet et al., 2017) or calibration approaches (e.g., Freyberg, 1988). This is commonly mentioned as a reason why hydrologists should be interested in machine learning models (e.g., Haaf et al., 2023; Nolte et al., 2023; Kratzert et al., 2019), as they may result in new knowledge that in turn may be used to improve empirical and process-based groundwater models. It is important, however, to be aware that model results are the combined result of the capabilities of the applied method and the choices made by the modeler in applying the method. These choices are often the result of personal judgment (Holländer et al., 2009) or experience of colleagues (Melsen, 2022), while some choices may simply be erroneous (Menard et al., 2021).

Several studies have compared models to simulate head time series (e.g., Sahoo and Jha, 2013; Shapoori et al., 2015; Wunsch et al., 2021; Zarafshan et al., 2023; Vonk et al., 2024). Many comparison studies, however, consider only one type of model (e.g., only statistical or process-based models), instead of different types of models. A fair and unbiased comparison of models is not straightforward, as a modeler may be more familiar with one model than another, or lack the experience to obtain optimal results for all models used in the comparison. A more fair and less biased comparison may be obtained by asking different modelers to simulate the same data with their model of choice. Simulations can subsequently be evaluated on independent (unseen) data. An example of this approach to model comparison is the karst modeling challenge (Jeannin et al., 2021) where different groups were asked to model a spring discharge time series. The present study was inspired by this project.

The *2022 groundwater time series modeling challenge* was organized by the first five authors of this paper, to compare the performance of different types of groundwater models in simulating hydraulic head time series under temperate, subarctic, and continental climates. The aim of the modeling challenge was to simulate the heads for a given $\sim$ 15-year calibration period and to predict the heads for an unknown $\sim$ 5-year validation period, with the validation itself performed solely by the organizers of the challenge. The fundamental idea was to investigate the strengths and weaknesses of different approaches, rather than to find the 'best' model to simulate heads. (Spoiler alert: there was no model that performed best on all performance criteria for all

head series.) Teams that wanted to participate were asked to model four time series of hydraulic heads from different regions with their model of choice. Seventeen teams took up the challenge and submitted results. An analysis of the results is presented in this paper. This paper is organized as follows. First, the setup of the challenge is presented, including the available data and performance metrics used in the evaluation. Next, the submissions are discussed and their performance is compared. The paper ends with recommendations and conclusions.

## 2 Setup of the challenge

### 2.1 Background

The groundwater modeling challenge was announced publicly at the General Assembly of the European Geophysical Union (EGU) in 2022 (Haaf et al., 2022), and further advertised via social media and personal communication. The challenge itself was administered through the GitHub platform (https://github.com/gwmodeling/challenge). On this website, all information and data were made public. All participants had access to the same information. There was no incentive given for participation in the challenge (e.g., there was no award for the best submission), other than potential co-authorship on this paper.

The modeling rules were kept relatively simple. Participants were provided with several time series of stresses (also called forcings, explanatory variables, or exogenous variables) that might cause the head fluctuations. The goal was to provide predictions of hydraulic heads for a validation period to the organizers; the organizers performed the validation. The participants were allowed to use any model and any or all of the provided stresses, with the only restriction that it was not allowed to use the observed head data itself as an explanatory variable (i.e., to predict the heads with historic head measurements). This restriction ensures that the models can be used to learn what stresses and processes result in the observed head dynamics (for example for use in a traditional process-based groundwater model), rather than the fact that the head at a certain time is strongly related to the head a few time steps prior. It was not allowed to use any other stresses than the ones provided, to ensure that any differences in the model outcomes were the result of the model and the provided stresses.

Participants were asked to submit the name of their team, their modeling results, and additional information about their modeling procedure using predefined submission formats. Submissions were required to include simulated heads, including uncertainty intervals, for both the period of observed heads and for an additional validation period for which stresses were supplied but no observed heads were made available. Further information on the submission had to be submitted through a README file in Markdown format. Finally, the modeling rules stated that "*The modeling workflow must be reproducible, preferably through the use of scripts, but otherwise described in enough detail to reproduce the results*".

### 2.2 Provided data

The participants were provided with four time series of hydraulic heads measured in monitoring wells as well as relevant stresses (e.g., climate data). Three wells are located in Europe (in the Netherlands, Germany, and Sweden), and one in the USA. The wells were selected to cover different hydrogeological settings (porous, fractured, and karstic aquifers, confined

and unconfined), different climates (e.g., influenced by snow or not) and other aspects (possible influence by surface water). Additionally, the organizers sought for long, mostly gapless times series with high-frequency head measurements (daily to weekly). It is noted here, however, that because this challenge only considered four wells, no statements can be made about which model performs best for which condition. The initial challenge contained measurements at a fifth monitoring well (named Sweden_1), but because none of the models could predict the measurements in the validation period (i.e., a Nash-Sutcliffe Efficiency below zero for all models in the validation period), it was removed from the challenge. Short descriptions of the settings of the monitoring wells and the general locations (country and region) were provided (similar to those found below) in addition to the head measurements and the time series of stresses that potentially affect the measured head fluctuations. Exact locations were not provided to guarantee that no team would be able to leverage the publicly available head data to predict the validation period. Detailed descriptions of the geology, setting, subsurface, and the construction of the monitoring wells were not provided to limit the challenge to use cases where detailed information about the subsurface is not available. This clearly limited the applicability of process-based models that generally require such information; no teams used such models as a result.

Observed hydraulic head data were provided for approximately 15 years for each monitoring well. The head data are of high-quality, with minimal gaps and (close to) constant frequency. Measurements were available approximately weekly for the well in Sweden, while daily data were available for all other wells. Head data with regular time intervals were selected to not exclude methods that cannot deal with irregular time series, even though these are often found in practice. Head data were selected from various continental and temperate climates and with seemingly negligible anthropogenic influences. The head series for each of the four wells are shown in Fig. 1, including the distribution of the measurements. A brief description of each monitoring well and its climatic setting is given in the following. The exact descriptions provided to the participants can be found in the Zenodo repository: https://zenodo.org/doi/10.5281/zenodo.10438290.

- The monitoring well in the Netherlands is located in the province of Drenthe in the northern part of the country. The monitoring well has the identification number B12C0274-001 and was downloaded from www.dinoloket.nl on September 15, 2022. The well is located in an unconfined aquifer, consisting of a ∼1.5 meter top-layer of peat material underlain by fine sands. The area is drained by many small ditches and drains. The water table regularly reaches the surface level. The surface elevation is about 11.33 meters above mean sea level, and the well is screened from 0.05 to 0.95 m below the surface in the peat layer. The climate is classified as a temperate oceanic climate, with an average annual precipitation of 876 mm and annual potential evaporation of 559 mm.

- The monitoring well in Germany is located in Bavaria in the southeastern part of the country and is drilled in the Upper Jurassic Malm Karst aquifer. It is a deep, confined aquifer (partially artesian), which is overlain by a local alluvial aquifer in a small river valley. The surface elevation is about 375 m above sea level, and the head is on average 0.9 m below the surface. The climate is classified as continental with warm summers. The average annual precipitation and potential evaporation are approximately equal, with 692 mm and 641 mm per year, respectively. The well has the ID 11149 and name Gungolfing 928. The head data was downloaded from https://www.gkd.bayern.de on May 5, 2022.

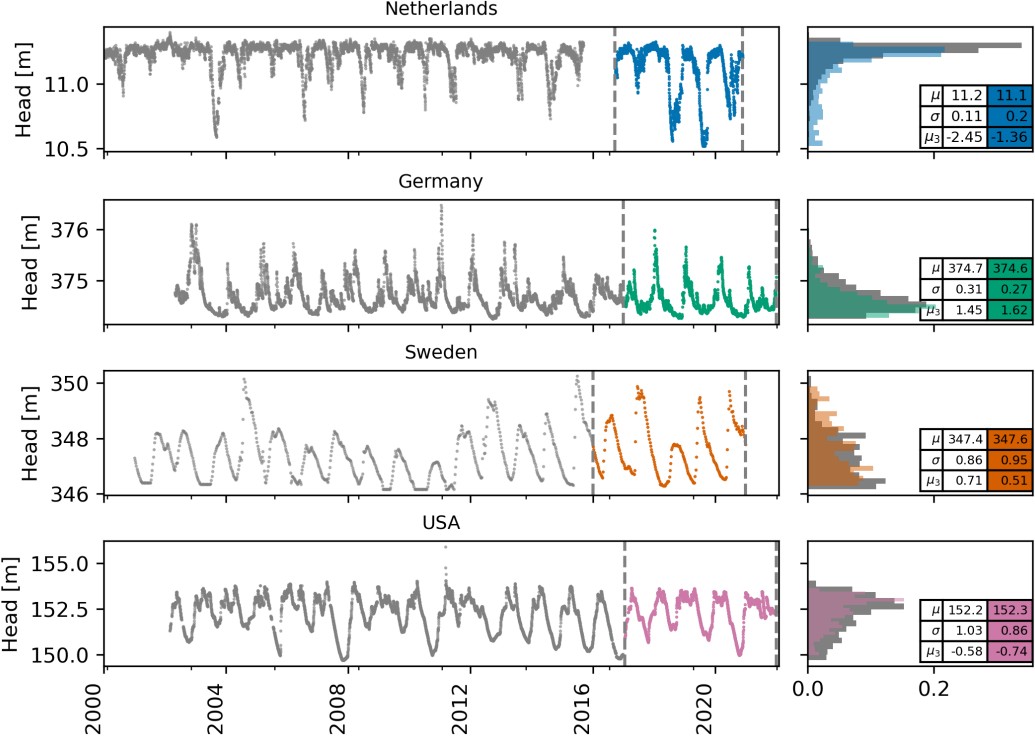

**Figure 1.** Hydraulic head time series, their probability density distributions, and summary statistics (the mean $\mu$, the standard deviation $\sigma$, and the skewness $\mu_3$) for the calibration period (gray data) and the validation period (colored data); heads in the validation period were not provided to the participants.

- The monitoring well in Sweden is located on a hillside in moraine terrain with shrub/grass-dominated land cover near Abisko National Park above the polar circle. The well is located in an unsorted till, containing both high contents of gravel and fine fractions with normal boulder frequency. The sediment thickness around the wells is between 5 and 10 m. The well is a steel standpipe with a perforated section of approximately 1 m and a diameter of 5.08 cm. The average distance to the groundwater table is 5.8 m and the length of the pipe is 7.4 m, reaching approximately to the fractured granitic bedrock. The climate is classified as subarctic, with cold summers and an average annual temperature around zero degrees Celsius. This location has the lowest amount of average annual precipitation (353 mm per year) and potential evaporation (332 mm per year). The head data was downloaded from https://www.sgu.se/grundvatten/grundvattennivaer/matstationer/ (site ID *Abisko 8*) on May 17, 2022 and processed using linear regression to fill short gaps and irregular measurement intervals.

- The monitoring well in the USA is located in the town Mansfield in the state of Connecticut and is screened in a confined bedrock aquifer. The aquifer consists of crystalline non-carbonated rock, predominantly metamorphic schist and gneiss, that is highly folded with numerous fractures and joints. The surface elevation is at approximately 157 m above sea

| Country | Lat. | Lon. | Climate | T | P | $ET_p$ |
|---|---|---|---|---|---|---|
|  |  |  |  | (°C) | (mm/year) | (mm/year) |
| **Netherlands** | 53.00 | 6.42 | Cfb | 9.9 | 876 | 559 |
| **Germany** | 48.92 | 11.35 | Dfb | 8.7 | 692 | 641 |
| **Sweden** | 68.36 | 18.82 | Dfc | 0.0 | 353 | 332 |
| **USA** | 41.81 | -72.29 | Dfb | 8.9 | 1344 | 956 |

**Table 1.** For each monitoring well: latitude (Lat.), longitude (Lon.), climate, and average annual values for the common meteorological stresses temperature (T), precipitation (P), and potential evaporation ($ET_p$)

level, and the well screen is approximately 135 m below the land surface. The well site number is 414831072173002; the head data was downloaded from https://waterdata.usgs.gov on July 1, 2022. The distance to the nearby river is approximately 1.5 km, with some small river branches even closer at around 500 m. The climate at this location is classified as continental, similar to Germany. The average annual precipitation and potential evaporation fluxes are, however, much larger, 1344 mm and 956 mm per year, respectively.

Time series of several stresses were provided to model the heads. All the variables were provided as time series with daily values starting approximately 10 years before the first head measurement and extending 4 to 5 years (the length of the validation period) after the last head measurement. For the wells located in Europe, stresses were obtained from the E-OBS database (Cornes et al., 2018, v0.25.0e). The data provided for these three wells included nine daily variables: precipitation, the mean, maximum, and minimum temperature, potential evaporation, mean sea level pressure, mean wind speed, mean relative humidity, and global radiation. For the well in the USA, only precipitation, potential evaporation (computed using the Hamon method (Hamon, 1961)), the mean and maximum daily temperature, and stage of a nearby river were provided; this was the only well where river stage data was provided. Summary statistics of the most common meteorological stresses (temperature, precipitation, and potential evaporation) are provided in Table 1 for reference and comparison.

### 2.3 Evaluation of modeling results

The aim of the modeling challenge was to simulate the heads for the calibration period and to predict the heads for the validation period. Therefore, the model evaluation is focused on how well the models predicted the heads in each of these periods. The simulated head time series and the estimated prediction intervals are evaluated and compared using various performance metrics, summarized in Table 2. All metrics are computed separately for the calibration and validation periods.

The simulated heads are evaluated using the Nash-Sutcliffe Efficiency (NSE), a relative error metric, and the Mean Absolute Error (MAE), an absolute error metric. In addition, it is evaluated how well the models simulate the lower and higher heads. The performance of the models to simulate low heads is measured using the MAE computed on the heads below the 0.2 quantile ($MAE_{0.2}$) while the performance to simulate high heads is evaluated using the MAE computed on the heads above the 0.8 quantile ($MAE_{0.8}$). The thresholds 0.2 and 0.8 are chosen based on a visual interpretation of the distribution of highs and

lows, but remains, of course, somewhat arbitrary. Analyses using other thresholds (0.05, 0.1, 0.9, and 0.95) showed that the results and conclusions about the models' performance to simulate the extremes does not substantially change with different
thresholds (see also the Supplementary Materials).

All teams were also asked to provide 95% prediction intervals for their simulations. The quality of these intervals is assessed by computing the Prediction Interval Coverage Probability (PICP). A PICP value of 0.95 means that 95% of the observed values are within the 95% prediction intervals. PICP values below 0.95 mean that the uncertainty is underestimated, while for PICP values above 0.95 the uncertainty is overestimated.

| Metric | Formula | Range |
|---|---|---|
| Nash-Sutcliffe Efficiency (NSE) | $1 - \frac{\sum_{i=1}^{N}(h_i - \hat{h_i})^2}{\sum_{i=1}^{N}(h_i - \mu_h)^2}$ | $-\infty$ - 1 |
| Mean absolute error (MAE) | $\frac{1}{N}\sum_{i=1}^{N}|h_i - \hat{h_i}|$ | $0$ - $\infty$ |
| Metric for low levels (MAE$_{0.2}$) | $\frac{1}{N}\sum_{i=1}^{N}|h_i - \hat{h_i}|$ for $h_i < h_{q=0.2}$ | $0$ - $\infty$ |
| Metric for high levels (MAE$_{0.8}$) | $\frac{1}{N}\sum_{i=1}^{N}|h_i - \hat{h_i}|$ for $h_i > h_{q=0.8}$ | $0$ - $\infty$ |
| Prediction Interval Coverage Probability (PICP) | $\frac{1}{N}\sum_{i=1}^{N} a_i, a_i = \begin{cases} 1 & \text{if } h_i \in [\hat{h_i}^L, \hat{h_i}^U], \\ 0 & \text{otherwise} \end{cases}$ | $0 - 1$ |

**Table 2.** The performance metrics used for model evaluation. $N$ is the number of measurements in the measured head time series $h_i$, $\hat{h_i}$ is the modeled head, $\mu_h$ is the average measured head, $h_{q=0.2}$ and $h_{q=0.8}$ are the 0.2 and 0.8 quantile of the measured head, respectively, and $\hat{h_i}^L$ and $\hat{h_i}^U$ are the lower and upper limits of the estimated prediction interval for measurement $i$, respectively.

## 3  Submissions

Seventeen teams from different institutes participated in the challenge. After an initial analysis of the results and consulting the participants, it was decided to exclude the results of two teams from further analysis. The models of these two teams had performance levels worse than if the mean head was taken as the simulation. Table 3 provides an overview of the data of the remaining fifteen participating groups. The geographical locations of participating groups are unevenly distributed, with
180 two-third of the teams coming from continental Europe. This was expected, as three of the four monitoring wells were located in Europe and the main promotion of the challenge was at the EGU. Twelve teams modeled all four time series provided for the challenge, and three teams simulated only two (for unknown reasons).

| Team | Model | Affiliations | Type | NLD | GER | SWE | USA |
|------|-------|-------------|------|-----|-----|-----|-----|
| da_collective | Pastas | 6, 7 | Lumped | ✔ | ✔ | ✔ | ✔ |
| gardenia | Gardenia | 8 | Lumped | ✔ | ✔ | ✔ | ✔ |
| HydroSight | Hydrosight | 9, 10, 11 | Lumped | ✔ | ✔ | ✔ | ✔ |
| Janis | RF | 12 | ML | ✔ | ✔ | ✔ | ✔ |
| Mirkwood | RF | 13 | ML | ✔ | ✔ | ✔ | ✔ |
| MxNl | RF/RBF-SVM/MLP/P-SVM | 15 | ML | ✔ | ✔ | ✔ | ✔ |
| Selina_Yang | SVR | 14 | ML | ✔ | ✔ | | |
| GEUS | LSTM | 16 | DL | ✔ | ✔ | ✔ | ✔ |
| LUHG | N-HiTS | 17 | DL | ✔ | ✔ | ✔ | ✔ |
| M2C_BRGM | BC-MODWT-DL | 18, 19 | DL | ✔ | ✔ | ✔ | ✔ |
| TUD | LSTM | 20 | DL | ✔ | | | ✔ |
| RouhaniEtAl | CNN | 21, 22 | DL | ✔ | ✔ | ✔ | ✔ |
| TUV | Transformer | 23, 24 | DL | ✔ | ✔ | ✔ | ✔ |
| haidro | LSTM | 25 | DL | ✔ | ✔ | ✔ | ✔ |
| uw | RNN | 26 | DL | ✔ | ✔ | | |

**Table 3.** Groups participating in the groundwater modeling challenge including the model they used, the type of model, and the wells that they simulated. ML=Machine Learning, DL=Deep Learning. Affiliation numbers correspond to the numbers on the first page of this paper.

All submissions were collected through the GitHub platform, where participating groups made pull requests to submit their results. The pull requests were manually checked before being merged. Many of the teams made sure that their analysis is repro-
185 ducible by submitting not only the results and a description of the models, but also the scripts and information on the computing environment used for the analysis. In general, the submissions were of high quality and the results were reproducible.

## 3.1 Model types

All teams used different models and software to simulate the hydraulic heads. The models were roughly categorized into three groups: lumped-parameter models, machine learning models, and deep learning models. None of the teams used a process-
190 based or analytical model, which may be explained (as mentioned before) by the limited description of the subsurface conditions and the exact locations of the wells. Detailed referenced descriptions of the individual models and methods can be found in the Supplement to this manuscript. The files submitted by the participants, including detailed scripts and workflows for most models can be found in a dedicated Zenodo repository (see section "Code and data availability"). An overview of the methods and the most important differences are provided below.

### 3.1.1 Lumped-parameter models

Three of the teams used a type of lumped-parameter model to simulate the heads. All three models (HydroSight (Peterson and Western, 2014), Pastas (Collenteur et al., 2019), and Gardenia (Thiéry, 2015)) use reservoir models to compute groundwater recharge from precipitation and potential evaporation. This recharge flux is translated into groundwater levels using a response function (Pastas and HydroSight) or a routing routine (Gardenia). Additional stresses can be added in a similar manner. These models are characterized by a small number of parameters (5 to 15) and short computation times.

Each team used different strategies for calibration. The parameters of the Pastas models were calibrated with PESTPP-IES and pyEMU (White et al., 2016). Parameters for the HydroSight models were calibrated using a shuffled complex algorithm (a global calibration scheme; Chu et al. (2011)). Gardenia was the only model in the challenge that was calibrated manually, by minimizing the NSE and through visual interpretation, even though the developers recommend to use Gardenia's automatic calibration procedure (Thiéry, 2024). All of these models used only precipitation and potential evaporation as explanatory variables, and some models used the temperature data (Sweden) and the river stage (USA) even though all three models have the ability to do both.

### 3.1.2 Machine learning models

Four teams used a supervised machine learning algorithm to model the heads: team Janis, Mirkwood, Selina_Yang, and MxNI. The first two teams applied (ensembles of) random forest models applying the R packages Tidymodels (Kuhn and Wickham, 2020) and 'ranger' (Wright and Ziegler, 2017). Both teams used the root mean squared error (RMSE) as the objective function. A detailed description of the random forest approach from team Mirkwood can be found in Di Ciacca et al. (2023). Team Selina_Yang used a Support Vector Regression (SVR) model from the Python package Scikit-learn (Pedregosa et al., 2011) to simulate the heads and used the mean squared error (MSE) as the calibration target. Team MxNI applied an ensemble of random forests, Multi-Layer Perceptron, Radial Basis Function support Vector Machine and Radial Basis Function support Vector Machine models in various configurations (Wright and Ziegler, 2017; Venables et al., 2002; Karatzoglou et al., 2004). The weights for each member in the ensemble were optimized using RMSE as the objective function.

All of these models used most of the supplied data, plus additional time series compiled from the supplied data; such compiled time series are referred to as engineered features. The features were often lagged versions of the original variables or moving window features (e.g., averages) to capture memory effects of the variables. Such engineered features can incorporate existing domain knowledge and thus provide the model with more relevant information, which can lead to better predictions and improved accuracy compared to leveraging the provided data directly.

### 3.1.3 Deep learning models

The majority of the teams, eight in total, applied a deep learning (DL) model to simulate the heads. Seven of them applied a model based on neural networks (e.g., CNN, RNN, LSTM). Four teams (TUD, Haidro, GEUS, and UW) used a Long Short-Term Memory (LSTM) network (e.g., Hochreiter and Schmidhuber, 1997), where Haidro used a multi-timescale LSTM

(MTS-LSTM). TUD used a sequence-to-sequence neural network model (Transformer, Vaswani et al., 2017), and Rouhani-EtAL applied a convolutional neural network (CNN, based on Wunsch et al., 2022). Team M2c_BRGM applied a Boundary Corrected-Maximal Overlap Wavelet Transform Deep Learning (BC-MODWT-DL) model based on (BI)LSTM and GRU models (Chidepudi et al., 2023). LUHG used a Neural Hierarchical Interpolation for Time Series Forecasting model (N-HiTS, Challu et al., 2022), optimized using a Gaussian likelihood function. All other teams in this group used the mean-squared error as the loss function to optimize the models. All these models simulate nonlinear relationships between the input data and the heads and used all the data provided for each well.

In contrast to the machine learning models, only three teams derived additional features from the provided input data (GEUS, LUHG, and Haidro). For example, team GEUS computed snow accumulation, to be used as an additional input variable. The reasons why other teams did not use feature engineering are not known, but although deep learning methods can benefit from additional features in a similar way to machine learning models (see above), they are better at extracting features themselves during training. In addition, the inclusion of additional inputs increases the complexity (number of parameters) of a deep learning model, which can be counterproductive.

## 4 Results

### 4.1 Overall performance

The overall model performances were measured by the Nash-Sutcliffe efficiency (NSE) and the mean absolute error (MAE), which are presented in Fig. 2 and Fig. 3, respectively. Each column shows a different monitoring well, and each row represents a team. The lighter colored bars show the metric values for the calibration period, and the darker colored bars for the validation period. The numbered circles denote the rank of each team per well in the validation period, and the star denotes the best model. At the top, box-plots of the metric values of all teams are shown. All box plots in this paper show the interquartile range and whiskers indicate 1.5 times the interquartile range. For the NSE, the highest value is best (Fig. 2). For the MAE, the lowest value is best (Fig. 3). Note that the MAE is expected to be higher for wells where the range of the head data is larger (i.e., the MAE is expected to be larger for the well in Sweden than for the well in the Netherlands).

Most models generally showed reasonable performance, except for the well in Sweden, with an average NSE of 0.64 over all four wells in the validation period. Without the well in Sweden, the average NSE for the three remaining wells in the validation period is 0.72. This shows the general capability of the models to simulate the observed groundwater levels with the provided input data. Model performances generally decreased from the calibration to the validation period, from a NSE of 0.80 to a NSE of 0.64 on average (for all four wells).

The results shown in Figs. 2 and 3 indicate that none of the models consistently outperformed all other models for all wells. Moreover, no models consistently ranked (based on these two metrics) in the top 5 for all 4 wells. There is also not one model type (Lumped, ML, or DL) that consistently outperforms the others. For the MAE metric, for example, the two best ranking models are deep learning models for the Netherlands, lumped-parameter models for Germany, and machine learning models for Sweden. The deep learning models, however, have the best models for two out of four wells, as measured by MAE and

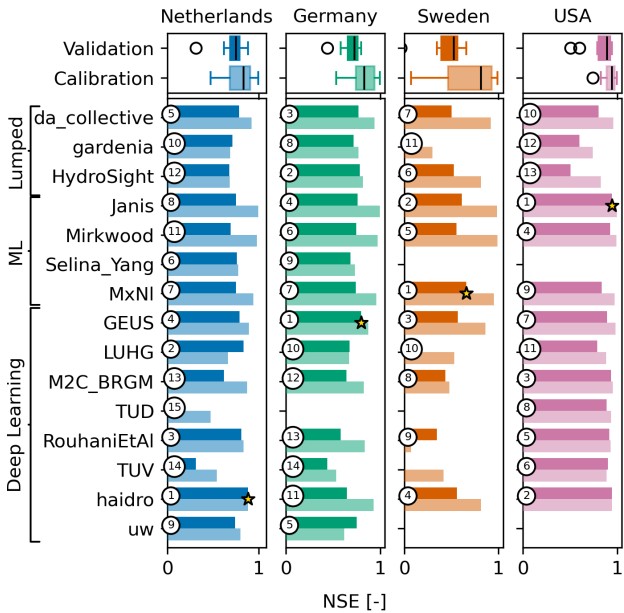

**Figure 2.** Bar plots of the Nash-Sutcliffe Efficiency (NSE) for each of the wells (each column) and team (each row), for the calibration (lighter color) and predicted validation period (darker color). The teams are grouped together by their model type. The numbered circles denote the rank of the team for each well, and the star denotes the best model per well in the validation period (highest NSE is best). If no model was provided by a team, there is no rank for that well. Box-plots of the metric values over all teams are shown for each well at the top.

NSE. The different performance metrics measure different error metrics (relative vs. absolute) or focus on different parts of the head time series (low heads or high heads). The ranking of the models differs for each error metric, which highlights the importance of using multiple metrics. Although the best-performing model is indicated for each metric, it must be pointed out that the difference between the top-performing models is small for each metric.

Performance of the lumped-parameter models is substantially lower for the well in the USA. This is also the only well where river stage data was provided. The relatively low model performances for HydroSight and Gardenia here can probably be explained by the fact that river stage data was not used in these models, opposite to all other teams. The Spearman correlation between the river stage and the head is R=0.78, making the river stage a good predictor of the head for this well. It is noted, however, that including the river as a stress is possible in both HydroSight and Gardenia, and would likely improve the results. The substantially lower performances may thus be considered the result of modeling choices rather than model deficiencies. Missing data and processes are likely also the reasons for the low model performance of the Gardenia model for the well in Sweden, i.e., it is the only model in the challenge that did not use temperature data, while Gardenia has an option to simulate the snow process. Temperature data for Sweden is important to account for the impact of snow processes on the heads.

The model fit is further illustrated by plots of the best model for each model type according to the NSE for each of the four wells (Fig. 4). The results show that the best models simulate the heads fairly accurately, except for the well in Sweden. For

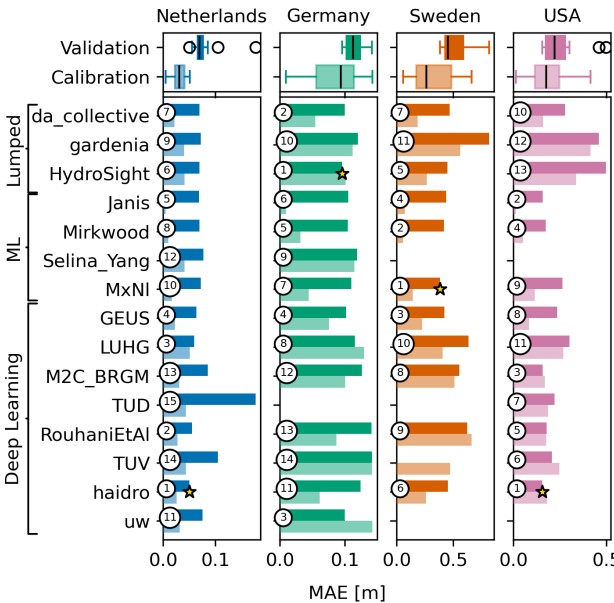

**Figure 3.** The same as for Figure 2 but for the Mean Absolute Error (MAE). Note that the lowest MAE is best.

the wells in the Netherlands, Germany, and Sweden, substantial differences exist between the three best models despite the use of similar or the same input data.

## 4.2 Low and high groundwater levels

The performance of the models to simulate low and high groundwater levels (lows and highs) was measured using the mean absolute error (MAE) computed on the head measurements below the 0.2 quantile and above the 0.8 quantile. The results for the $MAE_{0.2}$ and $MAE_{0.8}$ for the validation period are shown in Fig. 5. Again, there is no model that consistently outperforms the others for all wells on simulating low heads or high heads (or both).

In general, the MAE's are higher in the direction of the skewness of the head distribution (see Fig. 1). For example, the $MAE_{0.2}$ is higher than the $MAE_{0.8}$ for the Netherlands and the USA, which are skewed to the lower head values (i.e., the distribution has a longer tail for the lower heads), while the $MAE_{0.8}$ is higher than the $MAE_{0.2}$ for Germany and Sweden, which are skewed to the higher head values. For the data of the Netherlands in particular, where the heads are skewed due to capping near the drainage level, all models simulate the higher heads much better than the lower heads.

## 4.3 Comparison of uncertainty estimates

The teams were asked to provide uncertainty estimates in the form of 95% prediction intervals for the head simulations (one team didn't submit uncertainty estimates). It was tested what percentage of the measurements lay within the intervals using the PICP. Figure 6 shows bar plots of the PICP, where 0.95 is a perfect score. A lower value means the prediction interval is

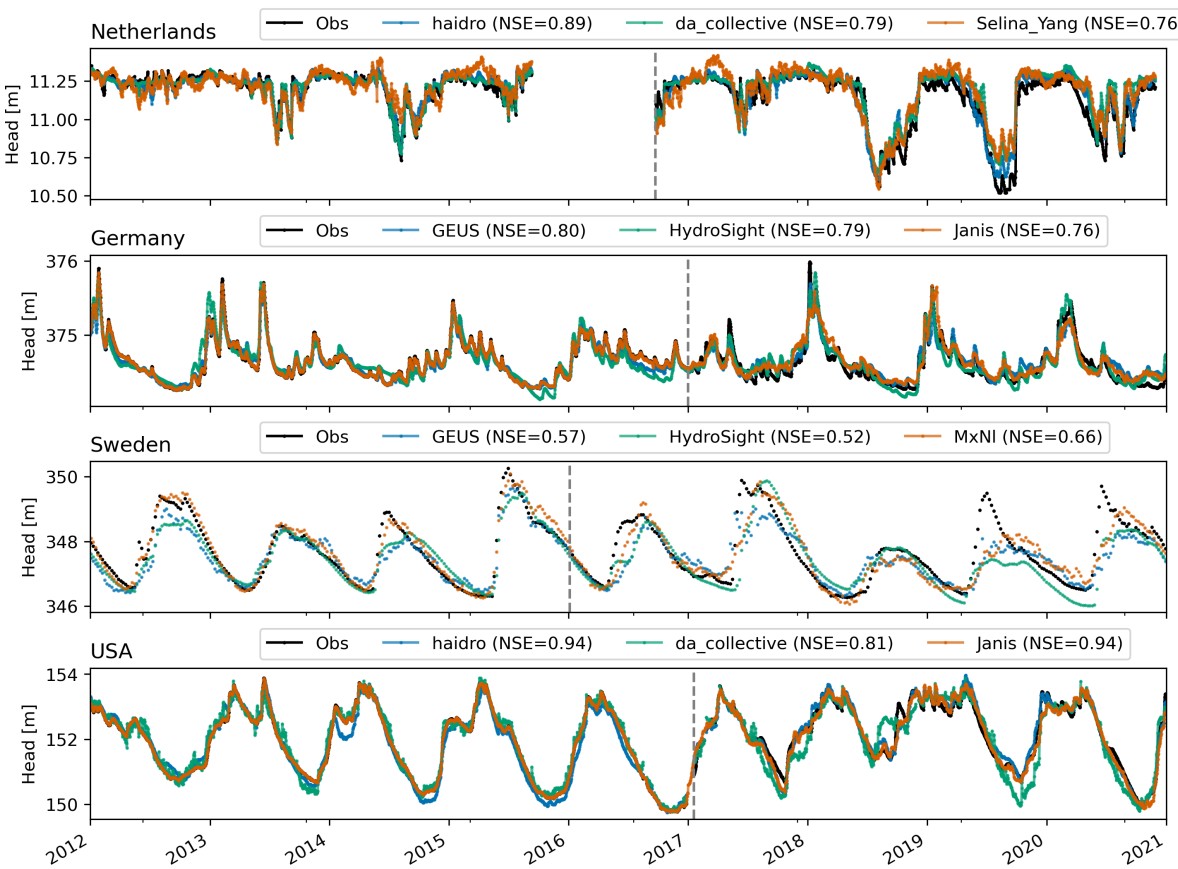

**Figure 4.** Observed and simulated head time series for the model with the highest NSE in the validation period (values shown in the legends) of each model type (blue line is deep learning, orange is lumped-parameter, and green is machine learning). The vertical dashed line shows the start of the validation period.

too narrow and a higher value means the prediction interval is too wide. It is noted that none of the models had a PICP of 1.0, which would indicate that the estimated 95% prediction interval is so wide that it includes all measurements. On the other hand, many teams severely underestimated the prediction intervals. Depending on the well, between 5 and 8 teams had a PICP of 0.5 or less, which means that at least 50% of the observed heads lie outside the 95% prediction interval.

Five teams, LUHG, GEUS, da_collective, HydroSight, and TUV had reasonable to very good estimates of the prediction intervals at all sites. These teams represent both lumped-parameter models and deep learning models, which indicates that the quality of the uncertainty estimates is not related to the type of model but to the method of uncertainty estimation. For the machine learning models, only Selina_Yang provided good estimates of the prediction intervals, but only supplied intervals for two sites. It is noted that some of the teams provided confidence intervals (indicated with a star * behind the team name in Fig. 300    6), which are narrower than prediction intervals.

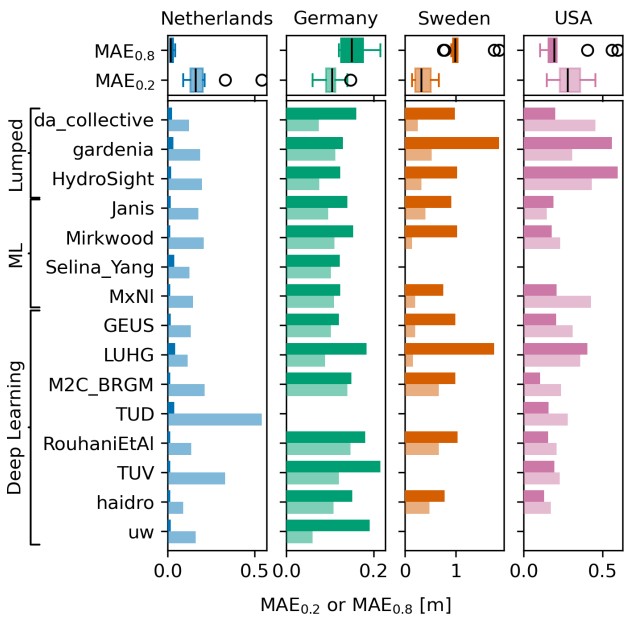

**Figure 5.** Bar plots of the $MAE_{0.2}$ (lighter color) and the $MAE_{0.8}$ (darker color) for the validation period (lower is better). Box plots of the metrics values are shown in the two rows at the top for comparison.

## 4.4 Evaluation of effort

All participating groups were asked to self-report estimates of the time investment required to develop the models and perform the calibration and uncertainty estimation. Figure 7 shows the time estimates, grouped into four categories: less than one hour, one to four hours, four to eight hours, and more than eight hours. The total time to develop and calibrate a single model ranged from less than one hour to nineteen hours, averaging approximately four hours per well. From the data shown in Fig. 7, it is clear that most models were developed and calibrated in less than a day. Some teams (at least team Janis, Team Selina_Yang, Team TUD) spent most of the time developing an approach for a single site and then used the same approach/script for all the other sites so it was much faster for the other sites. As a general rule, it appears that teams that estimated the uncertainty intervals more accurately (PICP near 0.95) had longer calibration times. While not surprising, it indicates that significant time investment is required to obtain good uncertainty estimates.

A straightforward interpretation of the time investments is difficult. One reason is that the actual computational time depends on the computational resources that were used. The estimates should thus be interpreted as an indication. Furthermore, the applied models and software may be at different states of development, and may have more options or only a few options to explore. Another less tractable reason is that the teams had varying familiarity with the used models: while some teams had previously used the model in one or multiple studies, other teams applied a model for the first time and started from scratch. Time investment may also be impacted by participants being at different stages in their careers, but no information to verify

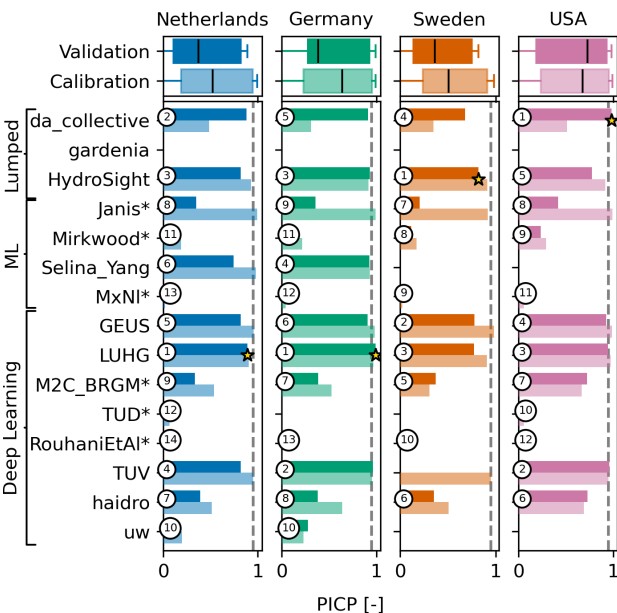

**Figure 6.** Bar plot of the PICP, as a measure of the quality of the prediction intervals (0.95 is best, indicated by the vertical dashed line). The star in the bar plot indicates the best model. A star behind the team name indicates that the team provided confidence intervals i.o. prediction intervals.

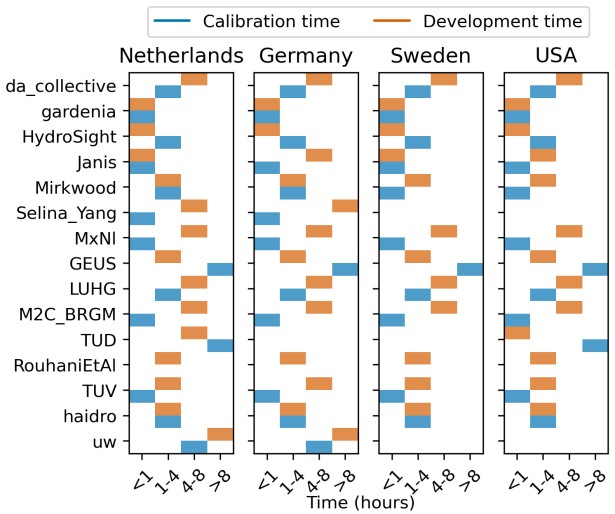

**Figure 7.** Estimated time of development (blue bars) and calibration (orange bars) in hours for each team.

this was collected. These problems suggest that model complexity cannot be assessed from time investment, but that other measures are needed to assess model complexity (e.g., Azmi et al., 2021; Weijs and Ruddell, 2020). Nonetheless, it can be

concluded that the time investment required to develop and calibrate data-driven models is relatively short compared to more classical groundwater models.

## 5  Discussion and Conclusions

Results were presented of the *2022 groundwater modeling challenge*. Seventeen teams picked up the challenge and submitted results (two of these were excluded after initial assessment of the results). The challenge focused on the simulation of four head time series with relatively limited information of the subsurface (a fifth series was removed from the challenge as none of the teams was able to obtain a NSE above zero in the validation period). All monitoring wells are located in areas with temperate or continental climates where the environment is more energy-limited rather than water-limited with relatively low inter-annual variability. Head time series were selected with (almost) regular time intervals not to exclude methods that cannot deal with irregular time series. The fifteen analyzed contributions were categorized into three general approaches: lumped-parameter modeling (three submissions), machine learning (four submissions), and deep learning (eight submissions). Performance was assessed using absolute and relative performance metrics for both the calibration and validation periods; the number of model parameters was not included in the performance metrics. The general dynamics of the measured hydraulic heads could be simulated reasonably well with at least one model from each category. No clear relationship between the model type and hydrogeological setting was observed. The general performance decreased slightly from the calibration to the validation period, as may be expected.

### 5.1  Learning from model comparisons

In general, the performance metrics for models that performed best in each category did not differ much. This means that each of the three major methods can, in principle, be applied to obtain reasonable results in the current challenge. Within the deep learning category, substantial variations were observed between the models: some were good while others were not (e.g., Fig. 3). Such variation was not observed in the other two categories, but those had substantially fewer submissions. This may reflect on the difficulty (some would say the art) of applying deep learning methods, but may also reflect on the experience of the different teams with the application of these methods; experienced teams generally obtain better results (e.g., Holländer et al., 2009; Melsen, 2022). The materials submitted by the teams may serve as a resource for others to further investigate and learn why these differences exist and what strategies work best.

The lumped-parameter models applied only stresses that are also applied in 'traditional' groundwater models: rainfall, potential evaporation, and river stages (and some used temperature for simulating snow processes) in combination with response functions with only a few parameters. If model performance is better when a stress is included, this does not necessarily mean that there is a causal relationship, as the stress can be a proxy of another stress that behaves in a similar manner. It is highly likely, however, that if a stress is needed to obtain good results in a lumped-parameter model and the stress makes physical sense, that a traditional groundwater model also needs this stress. The AI models generally used more of the provided (meteorological) data than the lumped-parameter models, including engineered data such as multi-day averages. The AI models

generally use many more parameters than the lumped-parameter models, which may result in overfitting when applied incorrectly. No overfitting was observed, however, as performance in the validation period was generally somewhat lower than in the calibration period, which is evidence of the aptitude of the AI teams. All models can potentially benefit from an improved estimate of the potential evaporation, which was estimated for all sites using a simple temperature-based method.

The performance increase of AI models was limited as compared to lumped-parameter models for most wells. This means that the lumped-parameter models, which are specifically developed to simulate head variations, reproduce the fundamental behavior of the groundwater flow systems included in the challenge reasonably well. Because some of the AI models learned more from the provided data (i.e., achieved higher performance), it is expected that small performance gains may be obtained for the lumped parameter models by improving some of the process representations (e.g., snow melt, evaporation) in the 360 models.

For the well in the USA, the AI models performed substantially better than the lumped-parameter models, which may be attributed to the fact that river stage data was not used in two out of three lumped-parameter models. The representation of the river stage in the AI models lead to better results compared to the one lumped-parameter model that used the river stage (da_collective), although it is noted that this team simulated the response to river stage variations as immediate rather than 365 using a response function to transform the stage to a groundwater response. In general, artificial intelligence approaches have more flexibility in the use and design of input data than lumped-parameter models, allowing many more types and sources of data, which may also be transformed and combined through feature engineering. Such transformations of input data, which can be interpreted as simple response functions that are further transformed by the AI model, have the primary aim to deliver information to the AI model that may lead to improved predictions. Such transformed input data increases flexibility and 370 improves the fit, but it is often not straightforward to interpret for hydrogeological system understanding. Therefore, AI models are generally characterized as black-box models. However, recent studies have shown that methods from the field of explainable AI (XAI) can be leveraged to gain hydrogeological system understanding from trained AI models (e.g., Wunsch et al., 2024; Haaf et al., 2023; Jung et al., 2024). Lumped-parameter models, on the other hand, rely on pre-defined (simplified) response functions that shed light on the mechanistic functioning of the hydrogeological system. As such, lumped-parameter models 375 may be used to unravel and quantify the effect of different stresses on the groundwater level over time, e.g., the effect of variations in rainfall vs. the effect of variations in river stage.

A common concern about machine and deep learning models is that they cannot accurately simulate hydrological extremes or extend predictions beyond observed conditions. Other model types (e.g., models constrained by empirical relationships) are sometimes thought to perform better in this regard. The results of the current challenge contradict this and indicate that no 380 substantial differences exist between artificial intelligence models on the one hand, and lumped-parameter models on the other hand, provided both are applied appropriately. This is in line with other analyses testing this concern for streamflow modeling (e.g., Frame et al., 2022). This was especially interesting for the Netherlands, where the validation period was much drier than the calibration period, resulting in much lower heads than in the calibration period (see Fig. 4), but yet, the best deep learning model performed better than the best lumped-parameter model in the validation period.

## 5.2 The impact of modeling choices on head simulations

The simulated heads are the combined result of the applied method and the choices made by the modeler(s). Many choices needed to be made by the modeler for each method, all affecting the final model outcome. The effect of (human) choices on the outcome of analyses is a known issue in the earth sciences (see e.g., Menard et al. (2021); Melsen (2022)), and other sciences (see e.g., Silberzahn et al. (2018)). It is difficult to distinguish between the effect of actual model deficiencies (i.e., a model's inherent inability to simulate certain hydrological behavior), and model choices (e.g., inclusion of a stress or the choice of the calibration procedure) due to the setup of this challenge. The effect of modeling choices in this challenge is apparent from the significant performance difference between teams that applied deep learning methods (Figures 2 and 3). Another good example is the effect of modeling choices made for the Gardenia model. During the public review process of this paper, the Gardenia developers pointed out that substantially better results could be obtained with Gardenia by choosing different model structures (e.g., including snow in Sweden and river stages in the USA) and using automatic calibration (Thiéry, 2024).

It is emphasized, however, that the results clearly show that the current setup and choices led to good results for most models and that even better results could have been obtained for many models if other modeling choices were made. This highlights how difficult model inter-comparisons are, and how important the choices from the modeler are for the final result.

## 5.3 Recommendations for future challenges

The organization of a modeling challenge is a challenge in itself. Five lessons learned from organizing this challenge are listed below, which are envisioned to be helpful recommendations for organizers of future challenges.

1. Clearly state the objective of the challenge and the methods of performance evaluation. The objective for this challenge was to predict the heads during both the calibration and validations periods, i.e., all aspects of the time series. The exact metrics used to evaluate the performance were not determined in advance. It is conceivable that teams would have reached different outcomes had they known the exact performance criteria beforehand. Further investigation into the effect of varying performance criteria during benchmarking is warranted, given that the paper demonstrated the significant influence of metrics on the final results.

2. Be as explicit as possible when describing the deliverables. This challenge, for example, initially asked for 'uncertainty intervals', although 'prediction intervals' were meant. As a result, some teams submitted confidence intervals, which could have been prevented with more explicit descriptions. Furthermore, the challenge did not explicitly ask for a description of the calibration method (global, local) nor a description of the method to estimate the uncertainty, which would have been interesting information.

3. Provide a clear structure for submission of information on data pre-processing. For example, teams applying AI models widely employed different engineered input data. Their design likely has an important influence on model performance in combination with model architecture. A clear structure for submitting this model information would have made evaluation of the effect of engineered input data more tractable across the different submissions and data sets.

4. Evaluate the response of the models to a few scenarios of future stresses. For example, for this challenge it would have been interesting to supply a few scenarios with a significant rainfall event or drought period in the validation period. A comparison of model results for such scenarios is pertinent information on how models behave, especially when performance is similar in the calibration and validation period.

5. Automate the validation of the submitted materials to save time and prevent errors. Continuous integration can possibly help to check new submissions, like unit tests for software development, especially when handled through pull requests on code-sharing platforms.

## 5.4 Concluding remarks

The *2022 groundwater time series modeling challenge* was a successful challenge, attracting many contributions from the groundwater modeling community. This allowed to compare a large number of different models, many of which can be successfully used to simulate head time series. The materials provided by all the teams participating in this challenge may be used to further explore different methods and learn from each other (see section Code and data availability).

Several of the teams that submitted results commented that although it took considerable effort to obtain good results and uncertainty estimates, the challenge did not explore the full potential of their model. It is emphasized that it is not possible to contribute below-average performance to either model deficiencies or model choices, due to the setup of this challenge. Modeling choices clearly affected the model results for this challenge (Thiéry, 2024) as it did for previous challenges (Holländer et al., 2009; Menard et al., 2021)). Some of the model choices that had significant impact on the model results include the choice or stresses (temperature or river stage), calibration method, response function, and engineered features. Especially the variation in the performance of the AI models highlights the importance of modeling choices. It is concluded that the setup of data-driven models takes relatively little time (i.e., hours), but getting good results still strongly depends on the choices made by the modeler.

This challenge investigates the performance of data-driven models for a small subset of conditions. New challenges may be organized for different climate zones with more inter-annual variations (e.g., longer dry periods or more arid climates), time series with significant gaps or missing data, or different purposes in groundwater modeling (e.g., drawdown estimation, recharge estimation), where other types of models (i.e., process-based models) are more commonly applied. One such challenge is already being organized to spatially predict nitrate concentrations (Nölscher et al., 2024).

*Code and data availability.* All data and code necessary to reproduce the results and figures shown in this paper are available from the Zenodo repository: https://zenodo.org/doi/10.5281/zenodo.10438290. There you will also find all the materials submitted by the teams to reproduce the modeling results.

*Author contributions.* The organizers team, consisting of the first 5 authors, set up the challenge. JS collected data for the well in the USA. All other authors submitted modeling results. The formal evaluation of the results was performed by RAC. RAC, EH, and MB wrote the manuscript, with comments and edits from TL and AW, and all other authors. Co-authorship does not necessarily mean that all authors agree with all statements made in this paper.

*Competing interests.* The authors declare no competing interests.

*Acknowledgements.* The organizers thank all the participating teams for joining the challenge. We thank the reviewers for their constructive comments that helped to improve the manuscript and the presentation of the results of this challenge. The work of team RouhaniEtAl was supported by OurMED PRIMA Program project funded by the European Union's Horizon 2020 research and innovation under grant agreement No. 2222. The contribution from team 'uw' has been supported by the Canada First Research Excellence Fund provided to the
455 Global Water Futures (GWF) Project.

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
