# Peer review of "Data-driven modeling of hydraulic head time series: results and lessons learned from the 2022 groundwater modeling challenge"

_Hydrology and Earth System Sciences, 2024_

## Community Comment (CC1)

**Preprint hess-2024-111**

**Data-driven modeling of hydraulic head time series: results and lessons learned from the 2022 groundwater modeling challenge**.

By Raoul A. Collenteur et al. 2024

raoul.collenteur@eawag.ch

This a very interesting paper about modeling groundwater hydraulic head time series.

While the results and conclusions are of significant interest, the modeling performed using BRGM's Gardenia computer code presents a clear concern.

It appears that for the presented project, the users of this computer code have not employed the recommended standard method for modeling hydrological time series. The following issues have been identified:

1. Manual calibration: The code's standard procedure involves automatic calibration. However, in this case, manual calibration was employed without any justification. It is not surprising that this deviation from the standard approach has resulted in inaccurate calibration.
2. Omission of snowmelt module: Even for basins in snow-dominated climates like Sweden, the snowmelt module was not utilized. Consequently, the obtained results are of poor quality.
3. Not using of double-reservoir schemes**:** Double-reservoir schemes are tailored for shallow water level time series, such as the "Netherlands" series. Their absence in this analysis has led to poor simulation of this time series.
4. Disregard of river level integration: The standard feature in the Gardenia computer code for integrating river stage series was not utilized. Using this feature would have significantly improved the results for the "USA" series.

The results presented, resulting from an inappropriate use, strongly discredit BRGM's Gardenia calculation code, which is unacceptable.

We independently modeled the four hydraulic head time series using the data provided in the appendix and achieved satisfactory results:

- In validation phase, the NSE coefficients obtained rank first or second for three out of four wells.
- The average validation NSE rank is 3.25, which is significantly better than the previously presented value of 10.25 (indicating poor performance).

We understand that the paper presents the results from the "2022 groundwater modeling challenge".

However this is our opinion, as having developed Gardenia computer code at BRGM, that this very interesting and valuable paper should be modified to correct the concern of the clear misuse of the model and of the clear discredit on Gardenia model.

Detailed comments:

**Line 22**: "for the well in the USA, where the lumped-parameter models did not use (or use to the full benefit) the provided river stage data"

Gardenia lumped-parameter model can integrate the provided river stage as an "external influence". Such an "external influence" is commonly used for the influence of nearby pumping, and also for the variation of river stage or river flow. Taking into account the river stage data for the USA well series significantly improved the NSE criterion during the calibration period: NSE was increased from 0.72 to 0.86

⇨ The sentence should be adapted. "most lumped-parameter models, except Gardenia, did not use…"

**Line 169:** "Gardenia was manually calibrated by minimizing the NSE and visual interpretation."

This not at all the correct way of using Gardenia. Gardenia, since its creation in 1977, is implemented with an automatic calibration method, the Rosenbrock algorithm. Gardenia is distributed with a tutorial of more than 20 examples, each one with automatic calibration. Gardenia has been used to model more the aquifer level (heads) or the river flow in more than 1000 sites. It has never been calibrated manually.

No wonder than calibrating manually the model leads to poor results.

⇨ Our simulations obtained with automatic calibration (computer time between 5 and 10 second for the calibration of each well) will be provided in attached files
⇨ The corresponding NSE and MAE criteria will be provided in attached files

**Figure 2:** Nash-Sutcliffe Efficiency (NSE). The bar plots and ranking of Gardenia do not at all reflect the results obtained with a normal use of the model.

Truly, this discredits this BRGM model (even if it mentioned, line 211 that "none of the models consistently outperformed all other models")

Indeed after a normal standard automatic calibration of the 4 wells on the calibration period, and then calculating the criteria on the validation period (where the observed heads were totally ignored during the calibration phase), we obtained very different results

Comparing our validation NSE to the NSE values (digitalized) from Figure 2:

Our Gardenia validation phase NSE:

| | | |
|---|---|---|
| Netherlands | validation NSE = 0.873 | => Rank = **1**, instead of rank 10; |
| Germany | validation NSE = 0.80 | => Rank = **1** (or 2), instead of rank 8 |
| Sweden | validation NSE = 0.611 | => Rank = **2**, instead of rank 11 |
| USA | validation NSE = 0.862 | => Rank = **9**, instead of rank 12 |

⇨ Average Gardenia rank = **3.25**, instead of rank 10.25 which would be fairly bad.
⇨ Gardenia rank = within the two best ranks for 3 wells out 4.
⇨ The true bar plot and ranks numbers should be corrected in Figure 2 (and in Figure 4)

**Figure 3:** Mean Absolute Error (MAE)

Comparing our validation MAE to the MAE values (digitalized) from Figure 2:

Our values of Validation MAE:
Netherlands = 0.057  => Approx rank = **3**, instead of rank 9,
Germany = 0.10        => Approx rank = **4**, instead of rank 10,
Sweden_2 = 0.383     => Approx rank = **2**, instead of rank 11,
USA = 0.255 =>          => Approx rank = **9**, instead of rank 12

- ⇨ Average Gardenia rank = 4.5, instead of rank 10.5 which would be fairly bad.
- ⇨ The true bar plot and ranks numbers must corrected in this Figure 3.

**Line 209**: "Model performances generally decreased from the calibration…"

Just for information: our Gardenia modeling: average NSE for the 4 basin:
Calibration 0.807, validation = 0.786 => Very small decrease.

**Line 220-224**: "Performance of the lumped-parameter models substantially lower for the well in the USA"

In the sentence "The relatively low model performances for HydroSight  here can probably be explained by the fact that river stage data was not used in these models, opposite to all other teams."

The 2 words "and Gardenia" should be deleted, as using the river stage for the simulation of the USA well, which is standard in Gardenia, yields a very high NSEs: 0. 862 => Rank = 3 for validation, and a very high calibration NSE = 0.893.

**Lines 223-226:**

"Missing data and processes are likely also the reasons for the low model performance of the Gardenia model for the well in Sweden, i.e., it is the only model in the challenge that did not use temperature data. Temperature data for Sweden is important to account for the impact of snow processes on the heads."

The sentence must be deleted. As a matter of fact, since about 1977 Gardenia is operational with a snow melting module. It make no sense to model a basin (or a well) subject every year to very long periods with negative temperature without using the standard snow melting module. (There are examples of this use in the tutorial provided with the code distribution).

(To our mind, in a lumped parameter model equipped with a snow melting module, disregarding temperature data in such a snow context is as inappropriate as disregarding potential evapotranspiration (PET) data or even precipitation data.)

Using the standard snow melt module, using temperature, for the Sweden_2 well yields satisfying NSEs: 0.611 => Rank = 2 for validation, and 0.777 for calibration.

---

## Author Response (AR1)

We thank all the reviewers for their constructive comments to our manuscript. Below, we reply to all the reviewer comments. The comments are in black and our replies are in green.

**Anonymous Referee #1**

The paper presents a modeling challenge performed by 15 teams from different institutions to reproduce temporal evolution of hydraulic heads at four monitoring wells, based provided meteorological data and a calibration time window with previously observed heads. The teams adopt different methods, with large predominance of methods based on artificial intelligence (AI). I find this experiment of much interest for the hydrology community, and particularly timely considering the increasing and widespread use of AI. For this reason, I think that the paper fits the quality standard of HESS and I recommend it for publication. I only have a few minor comments for the authors.

REPLY: Thank you for your kind words.

Comments to authors

Method description in section 3.1.1 to 3.1.3 could be slightly expanded to better highlight the difference between the different methods within the same category.

REPLY: We understand this comment. It is obviously not possible to describe all 15 models in detail, so we had to summarize. We added additional information in the paper (see section 3.1), and highlighted better that detailed model descriptions can be found in the Supplement to the paper (~0.5-1 page per participating team), and the scripts in the Zenodo repository (Lines 194-197). We particularly emphasize the use of engineered features in ML and DL models.

It is not sufficiently clear if any information about geology and setting (e.g., well depth) is provided to the teams.

REPLY: The description as presented in Section 2.2 (the four bullets describing the sites) is pretty much all the teams received regarding geology and setting. We added a sentence explaining this (Lines 119-120), including a reference to the github site where all the information was provided.

Minor issues:

L1 and L51: "2022 groundwater time series modeling challenge". I suggest to put this in italic or between quotes

REPLY: That is a good idea. Change made throughout the manuscript.

L70: "not allowed to use the observed head data itself as an explanatory variable." Could the authors develop more on this point? What kind of modeling would violate this rule?

REPLY: Here we are interested in investigating which stresses can explain the head variations, and if the head is used as a stress we don't learn about that. In machine learning it is rather common to use the measured variable itself to predict what happens in the future. This makes a lot of sense, as the head tomorrow is probably similar to the head today. And if there was an upward trend in the past month, then there may still be an upward trend in the next month. In traditional groundwater modeling (e.g., MODFLOW, FEFLOW, etcetera), this is not possible, of course, and stresses (forcings) on the model together with the physical principles that are in the model are supposed to create the correct behavior. We have added a few words to clarify to the reader what is meant here and why (Line 84-87).

L86: At this point of the reading, calibration and validation period have not been defined yet (except in the abstract), which might complicate the understanding of the sentence "calibrate the model without head measurements in the validation period". I suggest to reformulate this sentence.

REPLY: Good point. We modified the text throughout (i.e., Lines 8, 14, 63-65, 84) and made it consistent. We also don't use the word "forecast" anymore, and replaced this by "prediction" instead.

L96: is the descriptions in lines 97 to 121 the same that was provided to the participants?

REPLY: In essence, yes. But the wording was slightly different. We included a reference to the original wording (also in response to the earlier comment about geology and setting), see lines (119-120).

Figure 1: Authors should provide references for the head time series in the text or in the figure.

REPLY: We added the origin of the head time series to the text by adding the official ID and the URLs (see Lines 120-151).

Table 3: Team name is not indicative of the geographical provenience nor of the participants. A connection between participant names and group name should be provided. The acronyms ML and DL should be defined in the caption.

REPLY: We added a column with the number of the authors' affiliation(s) in Table 3. The geographical location of the participants is already provided in the authors' list, and it is mentioned in the text that two thirds of the teams come from continental Europe. The acronyms were added to the Table.

Fig. 2 and 3: Do the box plots use quartiles or 20%-80% quantiles?

REPLY: The box gives quartiles (25%-75%) and the whiskers are 1.5 times the interquartile range, as per default Matplotlib settings. We added to the text: "All box plots in this paper show the interquartile range and whiskers indicate 1.5 times the interquartile range" (Lines 253-254).

**Anonymous Referee #2**

This manuscript is very interesting by presenting the results from the 2022 groundwater modeling challenge. The provided data, evaluation of model results are well described.

REPLY: Thank you. Glad to hear you find the paper interesting and the results well described.

However, I suggest to reject the manuscript considering the following reasons:

(1) This manuscript is more like a summary (or technical report) of the groundwater modeling challenge. Scientific results and discussions is limited.

REPLY: Our paper is a scientific analysis and discussion of the challenge we organized. This is a very common format in the sciences; in hydrological research papers it is applied in, e.g., Jeannin et al., 2021, Holländer et al. 2009, or the "Battle"-series as part of the WDSA/CCWI conferences since the 1980's).

References:

- Jeannin, P.-Y., Artigue, G., Butscher, C., Chang, Y., Charlier, J.-B., Duran, L., Gill, L., Hartmann, A., Johannet, A., Jourde, H., Kavousi, A., Liesch, T., Liu, Y., Lüthi, M., Malard, A., Mazzilli, N., Pardo-Igúzquiza, E., Thiéry, D., Reimann, T., Schuler, P., Wöhling, T., and Wunsch, A.: Karst modelling challenge 1: Results of hydrological modelling, Journal of Hydrology, 600, 126508, https://doi.org/10.1016/j.jhydrol.2021.126508, 2021.
- Holländer, H. M., Blume, T., Bormann, H., Buytaert, W., Chirico, G. B., Exbrayat, J.-F., Gustafsson, D., Hölzel, H., Kraft, P., Stamm, C., Stoll, S., Blöschl, G., and Flühler, H.: Comparative predictions of discharge from an artificial catchment (Chicken Creek) using sparse data, Hydrology and Earth System Sciences, 13, 2069–2094, https://doi.org/10.5194/hess-13-2069-2009, 2009.
- Battle of Water Networks, 3rd International Joint Conference on Water Distribution Systems Analysis &Computing and Control for the Water Industry (WDSA/CCWI), Ferrara, Italy, https://wdsa-ccwi2024.it/battle-of-water-networks/ (last accessed July 8, 2024)

We added the reference from Holländer et al. (2009) to the paper as an additional example. We also added an important discussion point about modeling choices and model performance to the discussion (see also next response). We think our study adds to the literature, another, well documented, example of how choices from the modeller may affect model performance across all the different model types.

(2) The novelty of this work is limited both from the point of view of groundwater hydrology or the point of view of modeling. The machine learning models and deep learning models

conducted in the manuscript are all classic algorithms. Furthermore, numerical models are not included.

REPLY: The reviewer is correct that no new models are developed, but that is the whole purpose of a challenge: Use your own favorite model to try to model the data series provided in the challenge. The performance of these models to different sites, and the way the teams applied these models, gives insights into both the capabilities of the different models and the teams that applied these models. Traditional numerical models were not excluded, but were not submitted, as mentioned in the paper.

We added an important point to the manuscript on why such comparisons are important, but also difficult. We now discuss how modelling choices affect the results, troubling such inter-comparison studies (new section in 5.1, lines 395-408). This also relates to active research in the field on how choices affect model outcomes, and we added references to this part of the literature (see also lines (49-52).

(3) The details of the models (lump-parameters, machine learning, deep learning) are not illustrated which may because they all used the classic ones.

REPLY: We can obviously not describe the details of all 15 models in the manuscript, so we had to summarize, but we appreciate the desire for more details. All model input files are available on github to reproduce the results. We added additional descriptions and references (see changes document section 3.1), also in response to Reviewer #1, and clarified further that detailed descriptions for each model can be found in the Supplement (0.5-1 pages per team).

Anonymous Referee #3

The paper presents a collaboration effort between 15 Teams to compare the performance of different types of models to simulate groundwater heads at four boreholes. The paper is clearly written, and I think it is of interest for hydrogeological modelers.

REPLY: Thank you for your kind words.

I recommend the publication of this paper after addressing the following points:

In the introduction, the authors argue that modelling will increase our understanding of groundwater systems. Also, they mention that AI may results in new knowledge that may be used to improve empirical and process-based groundwater models. Unfortunately, the modelling outcome is not discussed within this context and the hydro-geological characteristics of the aquifer systems hosting these boreholes are not inferred from these models and not discussed in the paper.

REPLY: In the Introduction we state that "*Modeling makes such information explicit and increases our understanding of groundwater systems*" with reference to Shapiro and Day-Lewis (2022) who argue for reframing groundwater hydrology as a data-driven science. And we cite several authors that we can learn from, e.g., machine learning, to improve our groundwater models. Unfortunately, this does not mean, however, that it is possible to infer, e.g., the aquifer characteristics of the studied boreholes from the models. The purpose of the challenge was to predict the head variation for the four years beyond the measured (provided) head data (more explicitly stated in lines 64-65 now), and that is what the paper focuses on.

Information regarding the structures of the models and how these reflects the hydro-geological settings should be included. I expect that the lumped model structures and parameters to reflect the hydro-geological characteristics. If ML models are black boxes and nothing can be inferred from them, this should be explicitly mentioned in the paper and included in the discussion. It would be good to know the opinion of the Teams regarding the use of these models as it poses a philosophical question regarding their use especially in prediction mode.

REPLY: We judged that the final number of monitoring well (four) used in the manuscript would not be enough the link model structures to hydrogeological settings, as we now explicitly state in lines 101-102. We do argue, however, that we can learn about the stresses that may likely be required to explain the head variations, in lines 353-356.

ML models are indeed often referred to as black box models but that does not mean that we can not learn anything from it. There are examples in the context of explainable artificial intelligence (XAI) modeling of conceptual hydrogeological model interpretation, see for example, Wunsch et al. (2024), Haaf et al. (2023) and Jung et al. (2024)). This is added in lines 380-383.

The results of the challenge clearly indicate that ML models can perform just as well (or better) as the lumped-parameter models to simulate heads in the validation period. So there is an obvious use for ML models for the prediction of the heads. In the section 5.1, we tried to summarize what we learn from the comparison of the different models. We expanded this section to discuss how the modeling choices may affect the results, which may lead to suboptimal models, in a new paragraph (see lines 394-408).

References:

- Wunsch, A., Liesch, T., and Goldscheider, N.: Towards understanding the influence of seasons on low-groundwater periods based on explainable machine learning, Hydrology and Earth System Sciences, 28, 2167–2178, https://doi.org/10.5194/hess-28-2167-2024, 2024.
- Jung, H., Saynisch-Wagner, J., and Schulz, S.: Can eXplainable AI Offer a New Perspective for Groundwater Recharge Estimation?—Global-Scale Modeling Using Neural Network, Water Resources Research, 60, e2023WR036360, https://doi.org/10.1029/2023WR036360, 2024.
- Haaf, E., et al. (2023). "Data‐Driven Estimation of Groundwater Level Time‐Series at Unmonitored Sites Using Comparative Regional Analysis." Water Resources Research 59(7).

Please revise the text describing the data used for calibration and validation. Also, validation and prediction terms are used interchangeably. It is stated that 10 years of data including groundwater levels are provided for calibration, and five years without GWLs are provided for validation. Is this meant to be for prediction? But later it is clear that GWLs for the five years are used for validation. Is the validation done by someone other than the Teams after the submission of model output? Please clarify.

REPLY: We clarified the text (see lines 63-65), also in response to a comment from Reviewer 1. The validation was done by the organizers, which is now more clearly stated in lines 63-65 and 81-82.

For the USA site, it is mentioned that the nearest surface water is approximately 6.8 km. Later, it has been found that the river has an important role in improving the performance of the models at this site. What is the magnitude of the river stage fluctuations? Does the porous medium hydraulic characteristics justify the river control of GWLs at a borehole that is approximately 7 km away?

REPLY: We misstated that nearby surface water was 6.83 km away. That is the distance to the gauging station. The main river is only about 1.5 km away while the closest branches are around 500 m away. The paper was modified accordingly (see lines 149).

If possible, can you please explain why additional engineered input data are needed for ML and DL models and why these models are not able to self-adjust to avoid the need for this additional input?

REPLY: We appreciate that a large part of the readership likely consists of 'traditional' groundwater modelers that are relatively unfamiliar with data-driven models, so we added some additional sentences explaining this in the expanded section on Model Types (Lines 224-228).

Can you please rewrite or simplify the statement regarding the AI models and lumped model written in Section 5.1 Lines 302 and 306 as I find it confusing.

REPLY: We rewrote these paragraphs and clarified the text and hope this is now clearer (Lines 350-369).

**Anonymous Referee #4**

The paper presents the results of a benchmark study where different data driven models are applied to selected groundwater head time series. The study presents a discussion of the various results collected and reviews the whole procedure of the proposed challenge.

I find the paper interesting and I particularly appreciated the effort of sharing difficulties and lessons learned in the organization and management of this comparison study.

REPLY: Thank you.

However, there a few technical points that should be addressed, as listed below. Ultimately, in my view, the key weak point of this work is that it is hard to extract a take home message related to the model performances, that may be useful in the selection of an appropriate approach in another case not considered here.

REPLY: We added two points to the discussion and conclusion. First, that each contribution to the challenge is the combined result of the method and the modeling team. It is, obviously, not possible to contribute a below-average performance to a weakness of the method or a suboptimal application of the modeling team, and we stated this explicitly. See Lines 395-408 and 441-447.

Second, we included a discussion of the justification to include a stress into the model or not. In data-driven modeling, the justification comes from making one model with the river stage and one without the river stage, and comparing results. If the model performance is better with the stress included, this does not necessarily mean that there is a causal relationship, as the stress can be a proxy of another stress that behaves in a similar manner. We do argue, however, that if a stress is needed to get a good time series model, that it is highly likely that a traditional groundwater model also needs this stress. See lines 352-356

We further added our four main takeaways to the abstract of the manuscripts (lines 21-30).

I think addressing the following points would help in resolving this weakness.

1. The approaches compared likely differ in terms of parameterization, e.g. what is the number of calibration parameters to be optimized in the calibration phase? This information is not discussed properly, but is crucial for a fair comparison especially of calibration (training) performance. Model discrimination criteria could be used for these purposes (e.g., AIC, BIC, possibly KIC) to provide a fair comparison between models and the authors should be able to compute these metrics based on the provided materials.

REPLY: It is difficult to compare the number of parameters of a lumped-parameter model and a ML model because a lumped-parameter model uses on the order of 10 parameters, while a ML model may use more than 10,000 parameters. In general, using many

unnecessary parameters may result in overfitting. ML algorithms, when correctly applied, apply all kinds of safety guards to avoid overfitting. We tested whether overfitting was a problem by evaluating the performance of the models in the validation period. To clarify this in the paper, we added additional explanations about the large difference in the number of parameters and that no significant overfitting was observed (see Lines 358-363).

2. I understand the idea of assessing the models in the tails, however I find the two MAE (0.2 and 0.8) criteria quite crude and based on completely arbitrary thresholds that could influence the results. The authors should further elaborate on the robustness and significance of these two criteria.

REPLY: Every performance metric in itself is an arbitrary choice, but each metric represents a comparison between the observed and modeled series. We outlined our selection of performance metrics in Section 2.3. In this case the 20% and 80% quantile thresholds represent most periods with low and high water levels fairly well. Using other thresholds such as 5%/95% or 10%/90% exclude too many values from the evaluation for the series considered here. The latter thresholds usually originate from the surface water domain, where runoff curves have quite different characteristics and runoff peaks are more distinct. Therefore, we claim that the 20%/80% thresholds are a good measure for this specific evaluation. To support this reasoning, please check the figure below, which shows the different thresholds for the USA head data.

We elaborated on this aspect in the text to clarify our choice (lines 170-174) and, furthermore, computed the results for other thresholds (0.1, 0.05, 0.9, 0.95) to assess the robustness of these thresholds and share the results in the Supplemental Materials.

[Figure]

3. Details on the hydrogeological context were not shared with the participating teams, and I understand this is motivated by the objective of comparing approaches requiring information that can be widely available even for scarcely characterized sites. However, I think that in this discussion it would be beneficial to share the hydrogeological characteristics, even though these were not part of the challenge input data. How and why these particular wells were selected? Is there any results that can be further discussed when we jointly consider

hydrogeological data and model performance? This discussion would be beneficial to readers and could be useful for application of approaches similar to those presented here in other cases, where some hydrogeological data is actually available.

REPLY: The wells were selected on the one hand to reflect different hydrogeological settings (porous, fractured and karstic aquifers, confined/unconfined), different climates (e.g., influenced by snow or not) and other aspects (possible influence by surface water). On the other hand, we selected the wells by their available data (long and gapless time series of daily (weekly in the case of Sweden) heads). Though we tried to interpret the results of the models for each well and its specifics, four wells and 15 submissions are probably not enough to draw any robust general conclusions regarding hydrogeological data and model performance. For example, model type A generally performs better in hydrogeological setting X, while model type B is better in hydrogeological setting Y. We added our reasoning for selecting these series to the paper (Lines 98-102).

4. The conclusions of the study should be strengthened to include some technical discussion. Now they read a bit shallow and generic.

REPLY: We modified the abstract and conclusions to strengthen the most important messages of the paper (lines 21-31 and 441-448). We elaborated on the fact that the results of the teams are the combined result of the method and the modeling team (lines 394-408). A large part of the model performance is determined by how the model is set up and calibrated, as also shown by the community comment Of Dominique Thiéry. This highlights the fact that although it is generally relatively easy to set up a data-driven model, getting good results is still an art that is highly dependent on choices of the modeler.

Minor comment: the plot in Figure 4 for Netherlands has a strange behavior, I think due to an "empty" period between validation and calibration. Probably it would be better to leave this blank rather than linearly connect the two points.

REPLY: Good point. We had left it blank in Figure 1 and modified Figure 4 accordingly.

**Community Comment #1, Dominique Thiéry**

This a very interesting paper about modeling groundwater hydraulic head time series. While the results and conclusions are of significant interest, the modeling performed using BRGM's Gardenia computer code presents a clear concern.

It appears that for the presented project, the users of this computer code have not employed the recommended standard method for modeling hydrological time series. The following issues have been identified. These issues are described in details in the attached file.

It is our opinion that this very interesting and valuable paper should be modified to correct the concern of the clear misuse of the model and of the clear discredit on Gardenia model.

REPLY: We thank Dominique Thiéry for his comment and we understand his concern. It is the risk of software developers that people apply their model suboptimally, which may result in an undeserved bad reputation. We want to avoid that, of course. We unfortunately cannot include your results in the challenge after the fact, as you undoubtedly understand; everybody was invited to submit results, the validation data was made public when we submitted the paper, and we analyzed all submitted results.

We emphasized in the paper that modeling results are the combined result of the model and the modeling team and that it is entirely possible that other modeling teams can get better (or worse) results with the same method (lines 394-408). Hence, poor performance of a method for a certain site does not necessarily reflect a deficiency of the method (see also lines 49-52). We referenced this comment to the initial submission in the manuscript to explicitly refer the reader to potential improvements for the Gardenia model.

Comments the attached PDF:

The following issues have been identified:
1. Manual calibration: The code's standard procedure involves automatic calibration. However, in this case, manual calibration was employed without any justification. It is not surprising that this deviation from the standard approach has resulted in inaccurate calibration.
2. Omission of snowmelt module: Even for basins in snow-dominated climates like Sweden, the snowmelt module was not utilized. Consequently, the obtained results are of poor quality.
3. Not using of double-reservoir schemes: Double-reservoir schemes are tailored for shallow water level time series, such as the "Netherlands" series. Their absence in this analysis has led to poor simulation of this time series.
4. Disregard of river level integration: The standard feature in the Gardenia computer code for integrating river stage series was not utilized. Using this feature would have

significantly improved the results for the "USA" series. The results presented, resulting from an inappropriate use, strongly discredit BRGM's Gardenia calculation code, which is unacceptable.

REPLY: As explained above, model teams were free to use a method in whatever way they seemed fit. We cannot (and don't want to) enforce application according to standards set by the developers. It is the risk any software developer runs when making software available to the general public. We emphasize in the manuscript that manual calibration is not the method recommended by the code developers and refer to your HESS comment (lines 209-210). We further added explanations emphasizing the choices made not to include certain model components (see e.g., lines 211-212, 275-276, 279 and comments below).

We independently modeled the four hydraulic head time series using the data provided in the appendix and achieved satisfactory results: In validation phase, the NSE coefficients obtained rank first or second for three out of four wells. The average validation NSE rank is 3.25, which is significantly better than the previously presented value of 10.25 (indicating poor performance). We understand that the paper presents the results from the "2022 groundwater modeling challenge". However this is our opinion, as having developed Gardenia computer code at BRGM, that this very interesting and valuable paper should be modified to correct the concern of the clear misuse of the model and of the clear discredit on Gardenia model.

REPLY: We don't think "misuse" is an appropriate assessment. Looking at your results (after the challenge was over and the heads in the validation period were made available), a better assessment is "suboptimal". The results of team Gardenia are somewhere in the middle of the pack, e.g., scoring 8th place for the Germany data in Figure 2. We refer to our previous and following replies on how we handled your concern and highlight in the paper that the suboptimal performance is related to modeling choices and not the model.

Detailed comments:
Line 22: "for the well in the USA, where the lumped-parameter models did not use (or use to the full benefit) the provided river stage data" Gardenia lumped-parameter model can integrate the provided river stage as an "external influence". Such an "external influence" is commonly used for the influence of nearby pumping, and also for the variation of river stage or river flow. Taking into account the river stage data for the USA well series significantly improved the NSE criterion during the calibration period: NSE was increased from 0.72 to 0.86. The sentence should be adapted. "most lumped-parameter models, except Gardenia, did not use…"

REPLY: Gardenia was not the only lumped-parameter model that didn't include the river stage in the USA (also HydroSight), so we leave the sentence as is. We now stress in the

manuscript that including the river as a stress is an option for all lumped models and would (probably) improve the results (lines 274-276 and 403-405).

Line 169: "Gardenia was manually calibrated by minimizing the NSE and visual interpretation." This not at all the correct way of using Gardenia. Gardenia, since its creation in 1977, is implemented with an automatic calibration method, the Rosenbrock algorithm. Gardenia is distributed with a tutorial of more than 20 examples, each one with automatic calibration. Gardenia has been used to model more the aquifer level (heads) or the river flow in more than 1000 sites. It has never been calibrated manually. No wonder than calibrating manually the model leads to poor results. Our simulations obtained with automatic calibration (computer time between 5 and 10 second for the calibration of each well) will be provided in attached files The corresponding NSE and MAE criteria will be provided in attached files.

REPLY: Teams are free to choose their method of calibration. They were not (and can not be) forced to use a calibration procedure favored by the developers. For your information, and as an example, team "da_collective" also didn't use the built-in parameter estimation procedure in the Pastas software. We now emphasize in the manuscript that manual calibration is not the method recommended by the code developers and refer to your HESS comment (lines 209-210).

Figure 2: Nash-Sutcliffe Efficiency (NSE). The bar plots and ranking of Gardenia do not at all reflect the results obtained with a normal use of the model. Truly, this discredits this BRGM model (even if it mentioned, line 211 that "none of the models consistently outperformed all other models"). Indeed after a normal standard automatic calibration of the 4 wells on the calibration period, and then calculating the criteria on the validation period (where the observed heads were totally ignored during the calibration phase), we obtained very different results. Comparing our validation NSE to the NSE values (digitalized) from Figure 2: Our Gardenia validation phase NSE:
Netherlands validation NSE = 0.873 => Rank = 1, instead of rank 10;
Germany validation NSE = 0.80 => Rank = 1 (or 2), instead of rank 8
Sweden validation NSE = 0.611 => Rank = 2, instead of rank 11
USA validation NSE = 0.862 => Rank = 9, instead of rank 12
Average Gardenia rank = 3.25, instead of rank 10.25 which would be fairly bad.
Gardenia rank = within the two best ranks for 3 wells out 4.
 he true bar plot and ranks numbers should be corrected in Figure 2 (and in Figure 4)
Figure 3: Mean Absolute Error (MAE)
Comparing our validation MAE to the MAE values (digitalized) from Figure 2:
Our values of Validation MAE:
Netherlands = 0.057 => Approx rank = 3, instead of rank 9,
Germany = 0.10 => Approx rank = 4, instead of rank 10,
Sweden_2 = 0.383 => Approx rank = 2, instead of rank 11,

USA = 0.255 => => Approx rank = 9, instead of rank 12
Average Gardenia rank = 4.5, instead of rank 10.5 which would be fairly bad.
The true bar plot and ranks numbers must corrected in this Figure 3.
Line 209: "Model performances generally decreased from the calibration…"
Just for information: our Gardenia modeling: average NSE for the 4 basin:
Calibration 0.807, validation = 0.786 => Very small decrease.

REPLY:
As previously stated, and we are sure you understand, we cannot include your results to the challenge after the challenge was over and the validation data and results were published (in HESS Discussions); in retrospect it is really unfortunate you didn't submit your results during the challenge, as it would have been a worthwhile submission. We thank you for your clarification, which we now explicitly referred to in the paper (401-404) in the discussion section.

Line 220-224: "Performance of the lumped-parameter models substantially lower for the well in the USA" In the sentence "The relatively low model performances for HydroSight and Gardenia here can probably be explained by the fact that river stage data was not used in these models, opposite to all other teams." The 2 words "and Gardenia" should be deleted, as using the river stage for the simulation of the USA well, which is standard in Gardenia, yields a very high NSEs: 0. 862 => Rank = 3 for validation, and a very high calibration NSE = 0.893.

REPLY: Teams were free to decide which stresses to use in their models (as clearly stated in the paper). We explain here what a likely reason was for the under-performance of both teams HydroSight and Gardenia. We included that both HydroSight and Gardenia have the option to include river stages to emphasize that the lower performance was likely due to a modeling choice and not a weakness of the method (lines 275-276).

Lines 223-226:
"Missing data and processes are likely also the reasons for the low model performance of the Gardenia model for the well in Sweden, i.e., it is the only model in the challenge that did not use temperature data. Temperature data for Sweden is important to account for the impact of snow processes on the heads." The sentence must be deleted. As a matter of fact, since about 1977 Gardenia is operational with a snow melting module. It make no sense to model a basin (or a well) subject every year to very long periods with negative temperature without using the standard snow melting module. (There are examples of this use in the tutorial provided with the code distribution). (To our mind, in a lumped parameter model equipped with a snow melting module, disregarding temperature data in such a snow context is as inappropriate as disregarding potential evapotranspiration (PET) data or even precipitation data.) Using the standard snow melt module, using temperature, for the

Sweden_2 well yields satisfying NSEs: 0.611 => Rank = 2 for validation, and 0.777 for calibration.

REPLY: Again, teams were free to choose which stresses to include. We now explicitly stated here that Gardenia has the option to include snow (278-279), but team Gardenia chose not to use this option.